



# LARA: a Lagrangian Reanalysis based on ERA5 spanning from 1940 to 2023

Lucie Bakels[1], Michael Blaschek[1], Marina Dütsch[1], Andreas Plach[1], Vincent Lechner[1], Georg Brack[1], Leopold Haimberger[1], and Andreas Stohl[1]

[1]Department of Meteorology and Geophysics, University of Vienna, Vienna, Austria

**Correspondence:** Lucie Bakels (lucie.bakels@su.se)

**Abstract.** Meteorological reanalyses are crucial datasets in atmospheric research, providing the foundation for many scientific applications. However, most reanalyses follow a Eulerian framework, providing data at specific, fixed points in space and time. This fixed-location approach is suitable for many scientific analyses, but studies focused on transport in the atmosphere would benefit from a Lagrangian framework, which provides data along dynamic, continuous trajectories following the movement of air.

To achieve this, the Lagrangian particle dispersion model FLEXPART was driven off-line with data from ECMWF's (European Centre for Medium-Range Weather Forecasts) latest reanalysis, ERA5, to convert the Eulerian ERA5 data into a Lagrangian format. FLEXPART utilises the grid-scale winds from ERA5 and stochastic parameterisations of turbulence and convection to advect particles in a domain-filling mode, where the global atmosphere is represented by 6 million particles that move freely in the atmosphere, with their number density following closely the density of air. The resulting new Lagrangian Reanalysis (LARA) dataset has been stored in an easily searchable database and made accessible to researchers all over the world. It will enable a wide range of studies, including global and regional analyses of extreme events, water and energy transport in the atmosphere, and atmospheric energy budgets.

Here, we describe the data format, and how the data can be accessed and analysed. Using four examples, we give a non-exhaustive list of possible applications for which LARA could be used for. We show methods for how the evolution of air masses and their properties can be studied, and how climatologies can be established. Our examples include a study of the evolution of the Hadley cell circulation, a climatology of warm conveyor belt events, a measure of continentality by time it takes air to reach land from the ocean, and an evaluation of the dynamical consistency between subsequent ERA5 meteorological fields.

## 1 Introduction

In order to forecast the future state of the atmosphere with a numerical model that solves the prognostic equations of motion, the atmosphere's initial state must be known with high accuracy. To describe this initial state, meteorological observations are essential. However, these measurements contain errors, have incomplete spatiotemporal coverage, and an analysis of the initial state of the atmosphere based on observations alone (e.g., through simple interpolation) would contain dynamical in-





consistencies. Therefore, the observations are usually merged with a Eulerian model forecast from a previous analysis using sophisticated data assimilation methods (e.g. Courtier and Talagrand, 1987). The product of such a data assimilation system is a Eulerian representation of the atmosphere, in which the atmospheric state variables are provided at fixed locations in space and time. These analysis data represent our best estimate of the state of the atmosphere at the given time, ideally also including an estimate of uncertainty (e.g., through ensemble methods).

Time series of such analyses can be used for meteorological case studies. However, since weather forecast models change with time, long series of operational analyses are not well suited for the analysis of climate variability or trends, or the consistent comparison of different events. For such purposes, so-called reanalyses have been produced using models and data assimilation systems that are "frozen" over the reanalysis period, the longest of which span more than a millennium (Last Millenium Reanalysis by Hakim et al., 2016). Several agencies have produced reanalyses, including the Climate Forecast System Reanalysis (CFSR), created by the American National Centers for Environmental Prediction (NCEP) (Saha et al., 2010), the Modern-Era Retrospective analysis for Research and Applications (MERRA), created by the American National Aeronautics and Space Administration (NASA) (Rienecker et al., 2011), the Japanese 55-year Reanalysis (JRA-55), created by the Japanese Meteorological Agency (JMA) (Kobayashi et al., 2015), and the ECMWF Reanalysis v5 (ERA5), created by the European Centre for Medium-Range Weather Forecasts (ECMWF) (Hersbach et al., 2020). Notwithstanding the fact that also reanalyses contain inhomogeneities resulting from changes in the observation system (e.g. Sterl, 2004), they are an extremely rich source of information for scientists from many different disciplines and support a wide range of scientific inquiries. These datasets offer a comprehensive view of the atmosphere, integrating observations and model data to generate consistent and accurate historical records of meteorological conditions. Such reanalyses are indispensable for understanding weather patterns, climate variability, and atmospheric processes, and modern reanalyses also include ocean and land components.

Traditional reanalyses are all Eulerian, delivering data at fixed spatial locations. While this stationary framework is effective for many applications, it poses limitations for studies that require understanding the movement of air or its trace constituents. A Lagrangian translation, which tracks data along continuous trajectories (Stohl, 1998), addresses this gap. This approach is particularly beneficial for research focused on atmospheric transport processes and source-receptor relationships (SRR), where following the path of air parcels provides deeper insights into phenomena such as pollutant dispersion, greenhouse gas budgets, moisture and energy transport, or the structure of synoptic systems.

Studies that utilise the Lagrangian tracing of air parcels are plentiful, however, they are often restricted to case studies, or transport climatologies for specific sites or certain processes. For instance, Wernli and Davies (1997a) and Wernli et al. (2002) used trajectory calculations to explore dynamical aspects of cyclones; Stohl (2001) and Eckhardt et al. (2004) established climatologies of warm conveyor belts and other mid-latitude air streams; Dirmeyer and Brubaker (1999) studied evaporative moisture sources; Zschenderlein et al. (2019) examined processes determining heat waves; Stohl et al. (2003) and Sprenger and Wernli (2003) quantified stratosphere-troposphere exchange fluxes; and Stohl et al. (2002) characterized the pathways and time scales of intercontinental pollution transport. Each of these studies calculated their own set of Lagrangian trajectories for the specific period and region of interest to that study. This constitutes a considerable effort and requires specialized expertise, since it is necessary to produce or download the Eulerian meteorological input data and run the trajectory model before it is



possible to proceed with the analysis of interest. An alternative to this is to produce a global dataset of continuous trajectories that fill the entire atmosphere and are tracked over a long time period.

Stohl and James (2004, 2005) were the first to produce such a dataset, for the purpose of studying the atmospheric branch of the global water cycle. They discretised the entire atmosphere using air "particles" of equal mass and calculated their trajectories using FLEXPART (Stohl et al., 1998a) driven by ECMWF operational analyses. The original purpose of this methodology

was to diagnose atmospheric moisture transport by establishing a connection between regions of net evaporation and regions of net precipitation. They validated their method using an extreme precipitation event and identified the regions from which the precipitation originated. Several other studies have used a similar approach for quantifying source-sink relationships of water (e.g. Nieto et al., 2006; Gimeno et al., 2010; Chen et al., 2012). Läderach and Sodemann (2016) simulated the period between 1979 and 2013 using data from the ECMWF ERAInterim reanalysis and used a moisture source diagnostic (Sodemann et al.,

2008) to quantify the atmospheric residence time of water vapour. Their dataset is available on request. Reithmeier and Sausen (2002) calculated domain-filling trajectories online in a climate model, and Vázquez et al. (2024) produced a dataset of 30 million FLEXPART trajectories with a temporal resolution of 3 hours, based on the ECMWF ERA5 reanalysis for the period 1980-2023, available upon request. This dataset was produced with FLEXPART version 10.4 (Pisso et al., 2019). Since then, a new version of FLEXPART has been released, showing large decrease in particle distribution degradation in the troposphere

over time (Bakels et al., 2024). This new version also allows for the output of temporal-averaged properties, allowing for a more complete representation of the path of air through the atmosphere. Therefore, here we present an open-access general-purpose dataset of global domain-filling trajectories using these latest features of FLEXPART and covering the full period of 1940-2024 ERA5 reanalysis.

In this work, we present a global dataset that may be called a Lagrangian reanalysis (LARA). It provides the hourly positions

of six million particles whose trajectories were calculated using the latest FLEXPART version (11) (Bakels et al., 2024), driven by the ECMWF ERA5 reanalysis from 1940 to March 2024 (Hersbach et al., 2020). Meteorological data are provided at the particle locations, which can be used for a wide range of possible analyses. The dataset is open-access and is stored in ZARR data format (https://data.eodc.eu/collections/LARA). This paper describes the dataset production, its limitations, and gives four examples of possible applications. We demonstrate how the dataset can be used to trace air mass transport, how

to identify certain dynamical phenomena, such as warm conveyor belts, and how to diagnose properties of the underlying Eulerian reanalysis data. All analyses and plotting routines are provided in the supplement and can be used as templates for further explorations. A first application of a preliminary version of this dataset was presented by Baier et al. (2022) who used it to examine the transport of moisture and energy from the tropical Pacific and its relationship to the El Niño/Southern Oscillation (ENSO).



**Table 1.** List of ERA5 input variables used in the creation of the LARA dataset. Note that FLEXPART reads in more meteorological parameters, which are needed for simulating additional processes such as wet and dry deposition of chemical substances. However, these were not accounted for here and are thus not listed.

| Variable | Short name | unit |
|---|---|---|
| **3D fields** | | |
| U and V component of wind | u and v | m/s |
| Eta-coordinate vertical velocity | etadot | $s^{-1}$ (internally: $Pa\,s^{-1}$) |
| Temperature | t | K |
| Specific humidity | q | kg/kg |
| **2D fields** | | |
| U and V component of wind at 10m level | u10 and v10 | m/s |
| Surface pressure | sp | Pa |
| 2 metre temperature | 2t | K |
| 2 metre dew point | 2d | K |
| Sensible heat flux | sshf | $W\,m^{-2}$ (de-accumulated by flex_extract) |
| Eastward and Northward turbulent surface stress | ewss and nsss | $N\,m^{-2}\,s$ |
| Geopotential | z | $m^2 s^{-2}$ (converted to topography in m above sea-level by flex_extract) |
| Standard deviation of orography | sdor | m |

## 2 Methods

### 2.1 Input data

For the creation of the LARA dataset, we used the most recent reanalysis dataset of ECMWF, ERA5 (Hersbach et al., 2020), with hourly $0.5° \times 0.5°$ data as input on 137 vertical model levels, of which 88 are located below 20 km. ERA5 is a widely used global atmospheric reanalysis dataset that provides detailed hourly estimates of atmospheric, land, and oceanic variables, covering the period from 1940 to the present. ERA5 combines vast amounts of historical observational data with advanced modelling and data assimilation techniques, ensuring high temporal and spatial resolution and a high level of consistency. It is built on ECMWF's Integrated Forecast System (IFS) Cy41r2, using 4D-Var data assimilation to incorporate a wide range of observational data, including satellite observations and in-situ measurements. Table 1 lists the ERA5 input parameters used for producing our dataset. The ERA5 data were obtained from the ECMWF archives using the flex_extract software (Tipka et al., 2020).



## 2.2 FLEXPART

FLEXPART is a Lagrangian Particle Dispersion Model (LPDM) that advances particles by interpolating gridded meteorological data from ECMWF or NCEP to the particle positions and applying a numerical trajectory calculation scheme. The input data include grid-scale three-dimensional fields of wind velocities, temperature, specific humidity, as well as several surface fields (see Table 1). FLEXPART trajectories are based on interpolated gridded winds and stochastic parameterisations of sub-grid convection and turbulence. This means that particles move with the large-scale winds with stochastic motions superimposed. FLEXPART also uses the input meteorological data to calculate various derived variables, such as potential vorticity and the height of the tropopause. FLEXPART has a well-documented history (Stohl et al., 1998a, 2005; Pisso et al., 2019), and recently saw a new release by Bakels et al. (2024), which includes a live documentation that records any new additions and changes (https://flexpart.img.univie.ac.at/docs/). Therefore, we refer to Bakels et al. (2024) for a general description of FLEXPART's functionalities but highlight those additions that were specifically implemented in FLEXPART 11 for the creation of the LARA dataset.

Many improvements made in FLEXPART 11 were designed to address challenges encountered during multi-year simulations using the domain-filling option. For example, using the native $\eta$ coordinates instead of internally transforming all meteorological data to a vertical coordinate system in units of metres, reduced interpolation errors, and improved the compliance with the well-mixed criterion across most of the atmosphere (Thomson, 1987, Section 3.1). The initial distribution of particles within the domain-filling option was improved to better represent air density throughout the entire atmosphere. Furthermore, introducing OpenMP parallelisation decreased the computing time for our specific application by 75 % as compared to the earlier MPI parallelisation present in FLEXPART 10.4 (Pisso et al., 2019).

## 2.3 Set-up

For our purpose we used the domain-filling option, which makes it possible to discretise the whole atmospheric mass into a set of particles, which can be traced both forward and backward in time. Here, we traced the particles forward in time. We used 6 million particles, with equal mass, that are initially positioned in the full atmosphere such that the particle density is proportional to the air density. This means that each particle represents a fixed air mass, but depending on where it moves, its represented volume of air may change (larger volumes where the air density is lower).

The dataset spans the period of 1940 to 2023. Given that the full period from start to finish would take over a year to complete, and given the progressive particle distribution degradation that occurs during extended runs (Bakels et al., 2024) (see also section 3.1), we divided the full period in twelve streams of eight years, with an overlap of one full year for each period. The three-month overlap between the different computational streams guarantees that, in post-processing analyses, for every instance in time during the entire LARA period, particles can be traced both forward and backward in time, as long as the desired duration of tracking is shorter than 3 months.

The trajectories were calculated with a numerical integration time step of 300 seconds. We found that particle densities deviate by approximately 20% less from ERA5 air densities when using 300 seconds time steps compared to 600 second time





steps. A further reduction of the time step did not result in a level of improvement that would justify the increase in run-
time. However, to ensure correct particle behaviour, much shorter adaptive time steps were used in the turbulent atmospheric
boundary layer; 10% (1%) of the variable Lagrangian time scale for horizontal (vertical) motions. This corresponds to the
following FLEXPART options: `CTL=10` and `IFINE=10` (see Bakels et al., 2024, for details).

For the simulations, we used the Vienna Scientific Cluster 5 (VSC-5) AMD EPYC Milan compute nodes, which have
2×64 cores and 256 threads in total, and 512 GB RAM. We used the GNU Fortran compiler (GCC 12.2.0) with the fol-
lowing optimisation flags: `-O3` and `-march=native`. For the best use of non-uniform access (NUMA), we set `export
OMP_PLACES=cores` and `export OMP_PROC_BIND=true`. In total, running all 12 seven-year periods took approxi-
mately 290000 cpu-hours and a total of 32 days to complete.

## 2.4  The LARA dataset

The LARA data set consists of hourly particle output for the entire 84-year period (1940-2023). Table 2 lists the different
variables contained in the LARA dataset. Both instantaneous values at the output time as well as hourly values averaged along
the trajectories are available for their positions, potential vorticity, specific humidity, air density, temperature, and pressure.
Averaged values contain the mean of all instantaneous values produced at each integration time step (300 s) of the previous
hour. The atmospheric boundary layer height, tropopause height, and topography do not have a vertical dimension like the
previous listed fields. Therefore, these are saved as two-dimensional spatial gridded fields, with an extra time dimension for
the atmospheric boundary layer and tropopause height. If users want these variables at the particle positions, they need to
interpolate them.

As stated before, the dataset spans the period of 1940 to March 2024, and is divided into chunks of eight years, with an
overlap of one year. Data is initially written in NetCDF by FLEXPART, but converted and stored in monthly zarr files, organ-
ised as follows: `<PERIOD>`/`<YEAR>`/`<MONTH>`/`<VARIABLE>`. For example, the instantaneous temperature in January
of 1944 is located in `1940-1947/1944/01/T`. Zarr is a cloud-friendly data format implemented in Python that stores chun-
ked, compressed multi-dimensional arrays. Zarr offers better compressibility and access than NetCDF and is accessible using
Python's package xarray (Hoyer and Hamman, 2017) (at least since v2022.12.0). Using xarray and the default compressor
`Blosc(cname='lz4', clevel=5,shuffle,blocksize=0)` for zarr results in a size reduction of $\sim 15\%$ as op-
posed to using the default NetCDF `zlib` compressor. Opening and using zarr files with xarray is nearly identical to using
NetCDF files, but, for example, opening a month's worth of files normalised per variable is $\sim 60\%$ faster. Loading one day of
data for one variable into memory is $\sim 80\%$ faster using zarr than using NetCDF.



**Table 2.** List of the variables contained within the LARA dataset. The dimensions are defined at the top of the list, followed by a series of variables with the dimensions particle index and time. Lastly, meteorological data lacking a third spatial dimension is stored as a grid.

|  | Short name | unit |
|---|---|---|
| **Dimensions** | | |
| Particle index (1 – 6 million) | `particle` | index |
| Gregorian time | `time` | <YYYY-MM-DD>T<HH:MM:SS> |
| Gridded longitude (0,360,0.5) | `longitude` | degree east |
| Gridded latitude (-90,90,0.5) | `latitude` | degree north |
| **Particle data variables** (`particle`, `time`) | | |
| Longitude | `lon` | degree east |
| Longitude average | `lon_av` | degree east |
| Latitude | `lat` | degree north |
| Latitude average | `lat_av` | degree north |
| Height | `z` | metre above ground |
| Height average | `z_av` | metre above ground |
| Potential vorticity | `pv` | pvu |
| Potential vorticity average | `pv_av` | pvu |
| Specific humidity | `sh` | $\mathrm{kg\,kg^{-1}}$ |
| Specific humidity average | `sh_av` | $\mathrm{kg\,kg^{-1}}$ |
| Density | `rho` | $\mathrm{kg\,m^{-3}}$ |
| Density average | `rho_av` | $\mathrm{kg\,m^{-3}}$ |
| Temperature | `T` | K |
| Temperature average | `T_av` | K |
| Pressure | `prs` | Pa |
| Pressure average | `prs_av` | Pa |
| **Gridded data variables** (`latitude`, `longitude`, `time`) | | |
| Atmospheric boundary layer height | `hmix` | metre above ground |
| Tropopause height | `tro` | K |
| **Gridded data variable** (`latitude`, `longitude`) | | |
| Topography | `to` | metre above sea level |





The following Python code example shows how to retrieve one variable for one month and store it locally.

```
1: # using zarr version 2.9 or later, but lower than 3
2: import zarr
3: import xarray as xr
4: # open the remote dataset
5: ds = xr.open_zarr("https://data.eodc.eu/collections/LARA/2017-2024/2023/01/T")
6: # print
7: print(ds)
8: <xarray.Dataset>
9: Dimensions:    (particle: 5914326, time: 744)
10: Coordinates:
11: * particle   (particle) int32 1 2 3 4 5 ... 5914323 5914324 5914325 5914326
12: * time       (time) datetime64[ns] 2023-01-01 ... 2023-01-31T23:00:00
13: Data variables:
14: T          (particle, time) float32 ...
15: # example size: 18GB (one month)
16: # store locally (zarr, netcdf, pandas, ...)
17: ds.to_zarr("/path/on/your/machine/LARA-2017-2024-202301-T")
18: ds.to_netcdf("/path/on/your/machine/LARA-2017-2024-202301-T.nc")
```

## 3   Validation and limitations of LARA

In this section, we validate the dataset using the well-mixed condition (Section 3.1), and describe (potential) limitations in
using LARA (Section 3.2). Since ground truth trajectories are not available for the entire LARA period across all atmospheric
layers, it is not possible to make an absolute quantification of errors. However, using quasi-conservative properties (e.g. PV), it
is possible to show differences between methods, and to highlight additional errors made during the transition periods between
ERA5 12-hourly assimilation windows. Using the same parameters as those used to create LARA, section 3.2 of Bakels et al.
(2024) shows that in 2020, below 10 km, errors due to assimilation window transitions are negligible as compared to other
errors. Using the improved interpolation scheme of FLEXPART 11, a reduction of $\sim 15\%$ and $\sim 8\%$ in PV errors is observed
above 10 km and between 5-10 km, respectively. FLEXPART shows generally good agreement with tracer experiments (ETEX
and CAPTEX), with Spearman coefficients between 0.56-0.68.

### 3.1   Density distribution

The so-called well-mixed condition (Thomson, 1987) states that particles of a passive tracer that are initially well mixed in
position must remain so throughout the simulation. This is an important criterion to judge the quality of a Lagrangian model
simulation which, however, is often quite difficult to achieve in practice. While turbulent schemes have been developed to fulfill
this criterion well on short time scales in the boundary layer (including the turbulence scheme implemented in FLEXPART),

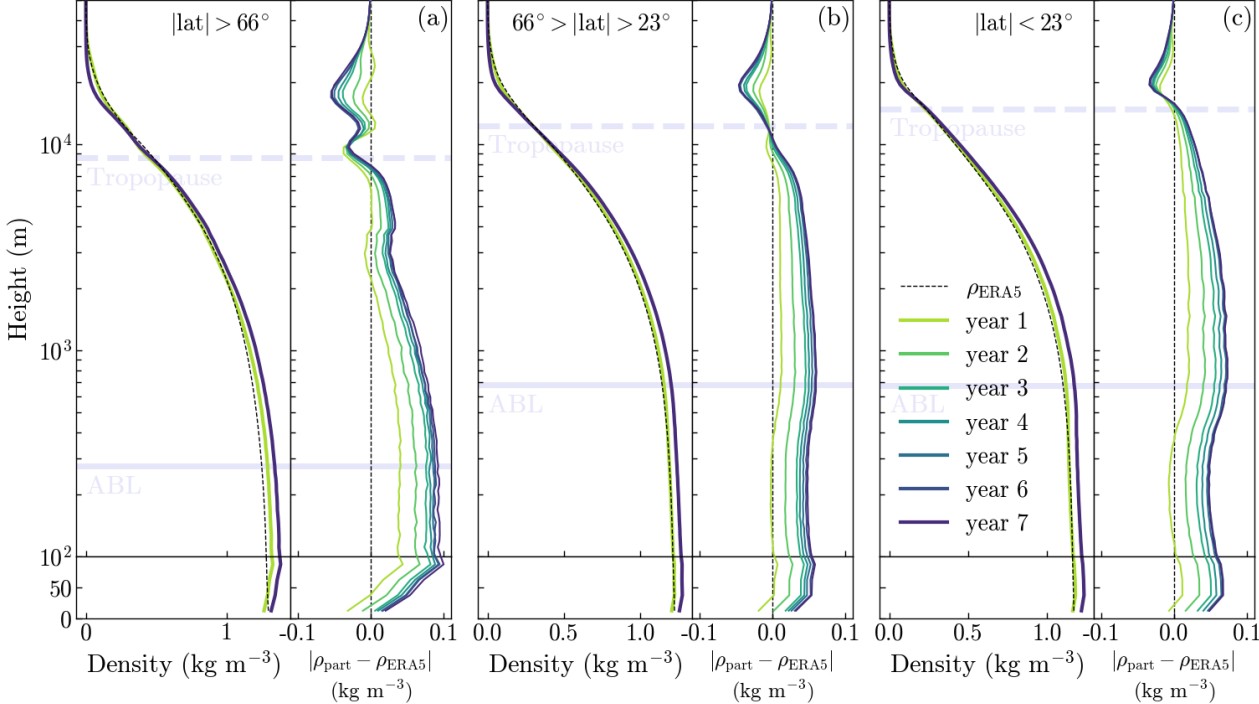

**Figure 1.** Density distribution of particles as compared to the density of air given by the ERA5 data. Each line represents the averaged density of every hour in the seventh month (July) of consecutive years after the start in January of the twelve simulations covering the 84-year period. Initially, the simulations start with a perfectly fitting particle density distribution that corresponds exactly to the air density profile. The results are separated into three regions: Polar regions (**a**), mid latitudes (**b**), and the tropics (**c**).

few global models were tested on long time scales. Deviations from well-mixedness can arise due to mass inconsistencies in the meteorological input data, interpolation errors, numerical errors, or errors in the convection or turbulence schemes.

Despite the modifications made to improve FLEXPART's adherence to the well-mixed condition, specifically the removal of an interpolation step by using the native $\eta$ coordinates within FLEXPART 11 (Bakels et al., 2024), as discussed in that study, and using a rather short maximum time step of 300 seconds when producing the LARA dataset, the condition was not entirely fulfilled. Following the method described in Cassiani et al. (2015), Fig. 1 shows the degradation of the vertical particle density profile as compared to the ERA5 air density profile within the LARA dataset over time. Our comparison is conducted

on a $0.5° \times 0.5°$ grid with 4 linear vertical levels below 100 m and 100 logarithmically spaced vertical levels between 100 m and 10 km. The different colours represent the mean July profiles, averaged over all twelve simulation streams, one to seven years after their start. We find similar patterns for all 7-year simulation streams and thus only show the average over the twelve simulation streams. The degradation rate is initially rapid, weakens after the third year and eventually stabilises around the fifth year. During the seventh year, below the tropopause we find height-weighted overdensities of about 3.5-6.6 % in our





data, which corresponds to an accumulation of particles in the troposphere; in the stratosphere, the sign is reversed (-65 %), indicating a lack of particles there. Tests showed that these deviations are not caused by FLEXPART's stochastic turbulence and convection parameterisations but must be due either to small mass inconsistencies in the grid-scale winds, their interpolation, or the numerical trajectory integration. The maximum deviation at any latitudinal-averaged volume is 0.6% of the particles that should be available in that given volume. This volume is located in the upper stratosphere. Despite this relatively low number,

there are no particle-free voids, indicating that particles fill the entire atmosphere over the full period of the simulations. This is important because it means that transport climatologies can be established from the LARA dataset for any location of the globe. However, a few words of caution are necessary, especially since deviations at discrete times can be larger.

Figure 1 shows that, when air density is calculated from the particle mass in a volume, there can be systematic deviations to the real air density. Therefore, it is always better to take the air density from the values interpolated to the particle positions,

not the mass of the particles. Similarly, when the total transported mass from a specific source or receptor volume is of interest, it may be preferable to avoid using the accumulated mass of the tracked particles. Instead, the mass can be calculated using the average air density within the source or receptor volume. Another potential issue is the use of the dataset for quantitatively studying transport processes on time scales longer than a few months, since there could be small systematic errors in calculated SRRs. This could affect, for instance, the calculation of the age-of-air in the stratosphere from multi-annual particle tracing

(e.g. Waugh and Hall, 2002).

## 3.2 Resolution limitations

For establishing reliable SRRs from the LARA dataset, a large enough number of particles is needed. A single particle cannot correctly characterise a transport pathway, due to the stochastic nature of turbulence and convection. The number of particles needed to obtain statistically significant results depends on the application. For example, to identify the transport pathway to a

specific location on a given date, it would be necessary to trace at least a few hundred particles. However, for characterizing the climate-mean transport pathway from or to a certain location, it is not necessary to trace this many particles at every time step. Since particle density is limited, this also means that the effective space-time resolution of the LARA dataset is limited and that always a certain search volume is needed, from which (or to which) particles can be tracked meaningfully. Since air and particle density decrease with height, larger search volumes are needed at higher altitudes than at lower altitudes. Therefore,

LARA is not suited to determine the dispersion of air from a point source, especially not for individual times. This means that a separate simulation might still be required to obtain the necessary high resolution. Nevertheless, a valuable addition to many such studies would be to have a climatology of the transport over extended periods of time. With our dataset this is possible.

## 4 Examples

In this section, we present a series of examples of potential applications of the LARA dataset. It should be noted that these

examples do not represent the full range of possibilities or are fully explored on themselves; however, they are sufficient for giving an understanding of the potential approaches that could be taken. In section 4.1, we demonstrate how particles



Earth System Open Access Science Data Discussions

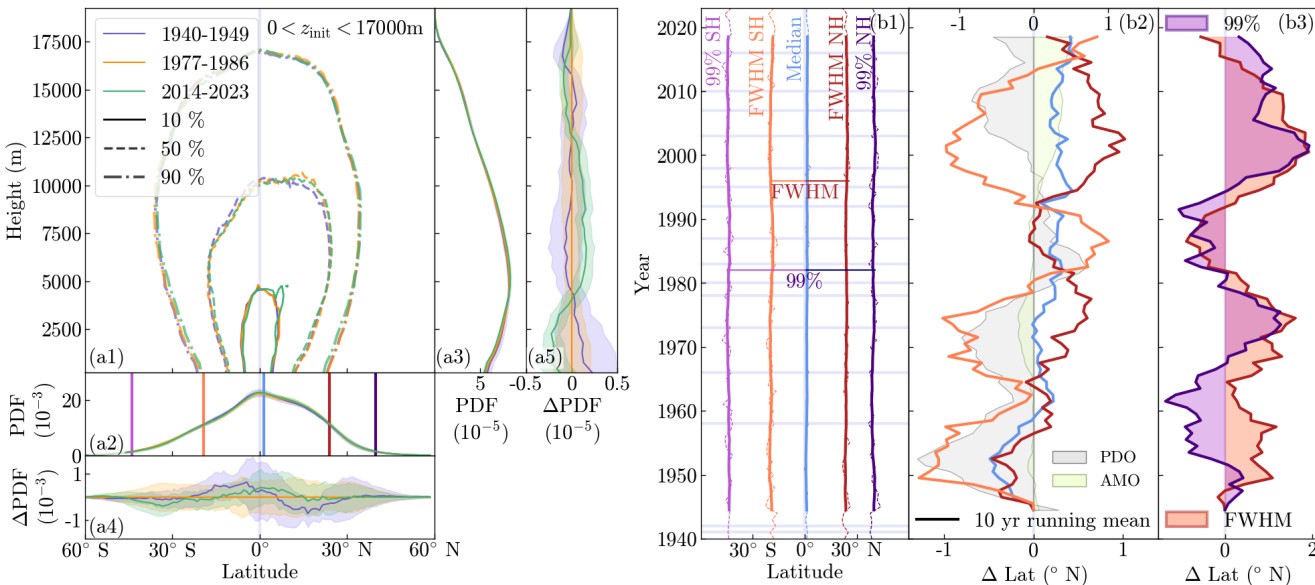

**Figure 2.** Hadley cell transport of air visualised by the position of particles 20 days after being located on the equator. Distribution of all particles initially positioned on the equator between 0 and 17 km altitude in March within the 10-year time periods 1940-1949 (purple), 1977-1986 (orange), and 2014-2023 (green). On the left, the distribution of all selected particles is plotted over latitude and height (panel **a1**). Side panels show the probability distribution function (PDF) of each period (**a2** and **a3**) and how the PDF of the latest and earliest period differ from the middle period (1977-1986) (**a4** and **a5**). Shaded areas denote the standard deviation based on yearly averages within each 10 year period. The solid vertical lines in panel **a2** show the median (blue), full-width-half-maximum (FWHM) on the southern hemisphere (orange), and northern hemisphere (red), and the 99th percentiles on the northern (indigo) and southern (pink) hemisphere averaged over the full 84 year period. Time series of the annual median, FWHM, and 99th percentiles are plotted (dashed lines), together with their 10 year running mean (solid lines), in panel **b1**. Horizontal lavender lines mark El Niño years. 10-year running mean differences in the location and the widths, respectively, of the median, FWHM and 99th percentile over time as compared to the mean of the first 10 years are plotted in the two right-most plots, **b2** and **b3**. The 10-year running mean of the Pacific Decadal and Atlantic Multidecadal Oscillations (PDO, AMO) indices for March and April are represented in panel **b2** by light gray and light blue patches, respectively, corresponding to the top axis.

and their properties can be selected and traced for an extended period of time. In section 4.2, we show how atmospheric phenomena can be identified by selecting particles based on their dynamical properties. In section 4.3 we present a Lagrangian definition of continentality by tracking the time particles spend over land. Finally, in section 4.4, we illustrate how LARA can
be employed as a diagnostic for testing the internal consistency of the ERA5 data. Our first two examples show trends that span the entire LARA period. However, since the quality of LARA is determined by its underlying ERA5 data, we urge caution when examining trends over periods that use different assimilation data.





## 4.1 Trends in equatorial transport

The response of the Hadley circulation to climate change has been studied extensively in recent decades, utilising a range of
methodologies, including reanalysis data (e.g. Stachnik and Schumacher, 2011; Zaplotnik et al., 2022), climate models (e.g.
Grise and Davis, 2020), and idealised general circulation models (e.g. Tandon et al., 2013). Figure 2 shows an example of how
LARA could aid in further understanding alterations in circulation patterns, exemplified by the Hadley cell. By monitoring
the distribution of particles originating from the equator over time, we can identify changes in the circulation properties of
the Hadley cell. In this example, we demonstrate one of LARA's fundamental capabilities, namely the quantification of mass
transport from selected regions. LARA is able to trace the movement of mass (and energy/moisture, etc.) forward to identify
downwind receptor regions, as well as locating upwind sources by tracing particles backward. We selected particles that crossed
the equator their first time during March, when the ITCZ is close to the equator, and traced them forward for 20 days. This was
carried out for each year within the full time span of LARA (1940-2023). The left panel of figure 2 shows the mean distributions
over latitude and height for three selected periods (1940-1949, 1977-1986, and 2014-2023), along with their respective standard
deviations, to provide insight into the temporal evolution of mass transport from the equator. For the most recent decade, the
vertical distribution of particles after 20 days shows a reduction within the lower parts of the lower atmosphere and an increase
at higher altitudes (>4 km), indicating an amplified rate of mass transport in the vertical direction.

As illustrated in panel (a2) by the vertical solid lines, we construct the annual median, full-width-half-maximum (FWHM),
and 99th percentile of the particle distribution in panels (b1), (b2), and (b3). The dashed lines in all panels represent a 10
year running mean, with the change over time relative to the earliest twenty-year mean plotted in panel (b2) and (b3). As
visible in panel (b2), the median position of the particles selected in March shifts northwards with time. This could be related
to the widening of the Hadley cell in recent decades, which has occurred primarily due to its northward extension in the
Northern Hemisphere (Studholme and Gulev, 2018). A higher value of FWHM equates to a circulation pattern that has a
broader distribution with a less pronounced peak. Therefore, we use the FWHM as a measure of Hadley cell broadening.
Although panel (b3) shows a wider FWHM during later periods (>1990) in comparison to earlier periods (<1990), the results
do not show a progressive broadening trend. Instead, they indicate a cyclical variation. This cyclic pattern may be attributed
to the Atlantic Multidecadal Oscillation (AMO), as proposed by Zaplotnik et al. (2022), who found a correlation between
a strong AMO and a narrowing of the Hadley cell (i.e. stronger Hadley circulation). The AMO index, provided by https:
//psl.noaa.gov/data/timeseries/AMO/ (Enfield et al., 2001) based on the NOAA ERSSTV5, only shows a weak relation in
the period before 1990, where a negative AMO index roughly corresponds with the broadening of our FWHM (broader and
therefore a weaker Hadley circulation). However, there is no such relation visible after the year 1990, where we see a stronger
AMO index and a broadening of the Hadley circulation. In contrast, a much more similar pattern over the whole 84 year period
can be found in the Pacific Decadal Oscillation (PDO), as provided by https://www.ncei.noaa.gov/access/monitoring/pdo/
(Zhang et al., 1997). Its 10-year running mean shows local peaks around 1952 (minimum), 1971 (minimum), 1982 (maximum)
and 2009 (minimum), at similar times of local maxima and minima we observe in the FWHM for the first three extremes. The
relation seems to break down around the year 1990, with the last PDO index minimum taking place almost ten years after the





local FWHM minimum. This underlines the fact that the Hadley cell circulation is a complicated process in which many factors could be causing anomalies and cannot be solely explained by an index such as the PDO. Therefore, to accurately determine the origin of the observed patterns, it is necessary to divide the globe into regions (i.e. Pacific, Atlantic, Indian Ocean). Only

then can other processes be investigated using LARA, e.g. global warming, which has been demonstrated in climate models to result in a broadening of the Hadley cell (e.g. Grise and Davis, 2020).

One of these factors causing anomalies are ENSO events. We illustrate El Niño years by lavender horizontal lines in panel **(b1)**. If we isolate El Niño years and compute their FWHM, these deviate from the values shown in panel **(b3)** by only -0.2 degrees on average. However, although following the same trend as the 99th percentile width as shown in **(b3)**, their 99th

percentile width is significantly narrower with an average of 2.2 degrees as compared to non-El Niño years. This is in agreement with studies such as Hu et al. (2018), which report a narrowing of the Hadley circulation during El Niño events.

This example presents an approach that can be readily extended. As we concluded above, it is important to decompose the analysis into different regions. Another example would be to repeat this example for each month of the year, selecting particles along the ITCZ. This could be used to study differences in seasonality, or changing circulations and pathways could be

examined by considering the three-dimensional trajectories. Furthermore, it is possible to trace the properties of the transported air, such as its moist static energy and specific humidity.

## 4.2 Warm conveyor belt events

In this second example, we demonstrate the strength of LARA in using particle dynamics to identify atmospheric phenomena, in this case warm conveyor belts (WCBs). Warm conveyor belts are streams of moist, warm air that ascend ahead of cold fronts

in extratropical cyclones, contributing to cloud formation and precipitation. In line with the approach initially taken by Stohl (2001) and Eckhardt et al. (2004), who used trajectory calculations to construct a climatology of WCBs for the 1979-1993 period, we sought to create two 41-year climatologies of WCBs for the periods 1940-1980 and 1981-2021, with a particular focus on the differences in occurrence between the two periods. We adopted one of the selection criteria for finding WCBs as used by Madonna et al. (2014), who studied the period between 1979 and 2010. This criterion states that particles must start at a

level below 790 hPa and ascend by at least 500 hPa within a two-day period. The model used in this study, LAGRANTO (Wernli and Davies, 1997b), does not parameterise convection. Consequently, particles can only rise by large-scale vertical motions. In contrast, in FLEXPART particles can also be lifted by the convection parameterisation, which would lead to erroneous classification of deep convection as WCBs when using only the ascent criterion by Madonna et al. (2014). To prevent this, we also apply the criterion by Eckhardt et al. (2004) that particles need to travel at least 10° eastward and 5° poleward, which

excludes most tropical convection. We also require the air mass within a WCB per square metre to be larger than 1 kg, implying a certain degree of coherence in airstream motion. However, this did not remove all instances of tropical convection, as visible in figure 3. We therefore note that the selection criterion could be improved upon in potential future work.

Figure 3 reveals similar overall patterns in the different seasons to those observed in earlier studies. They show maxima over the North Pacific and western North Atlantic during NH winter, and east of South America and over the Southern Pacific

during NH summer (e.g. Eckhardt et al., 2004; Madonna et al., 2014). We did not identify any WCB events at the foot of





**Figure 3.** Warm conveyor belt (WCB) climatology. Hourly mean air mass located within WCBs for the months of June, July, and August (green) and December, January, and February (purple)(**a** - dashed lines). 10 year running means and total means are represented by solid and dotted lines, respectively. Horizontal lavender lines mark El Niño years. The middle panels (**b,c**) show the mean hourly air mass located within a WCB for the period between 1981 and 2021, mapped onto the WCB starting points, for JJA (b) and DJF (c). The bottom panels (**d,e**) show the absolute difference between the period 1940-1980, and the period 1981-2021, for JJA (d) and DJF (d). Blue shades indicate a higher WCB incidence during the first 40 years (1940-1980), whereas red shades denotes a higher WCB incidence over the later 40 year period. Particles are selected on a two-day interval basis. All particles that started at a level below 790 hPa and ascended to a level above 500 hPa within the two-day interval are selected. Selected air masses are plotted at the beginning of the two-day selection period (i.e., WCB starting points), mapped on a 0.5° by 0.5° grid, divided by the area surface of their grid cells and averaged over the total number of days within the selection period. A lower limit of 1 kg m$^{-2}$ h$^{-1}$ is set.





the Himalaya during the NH summer, as observed in Madonna et al. (2014). This finding aligns with the results reported by Eckhardt et al. (2004). Madonna et al. (2014) hypothesised that the cause of this discrepancy was the Eckhardt et al. (2004) criterion of trajectories having to move at least 10° east and 5° north. However, the removal of this criterion did not result in distinctly different patterns in the north of India. We therefore hypothesise that differences in trajectory models, particularly in
the treatment of topography, are responsible for this discrepancy.

The global total mass within WCBs shows an increase between around 1970 until 1995, whereas we see more or less stable values before and after. The differences between the selected periods separated by 1981, show a broadening of the starting point distribution of WCB events in the more recent period over the South Pacific Ocean during NH summer. An increase of WCB events in the later period seems to occur over the South Atlantic and the Indian Ocean during the NH summer. During
the NH winter, we also observe an increase of WCB events in the North Atlantic and North Pacific. However, this appears to be accompanied by a slight north-western shift. During the same period, a slight increase is visible in the western Mediterranean Sea, while a minor decline is discernible in the eastern region. This has the potential to impact precipitation during intense cyclones, as evidenced by Flaounas et al. (2018), whose findings indicate that WCBs contribute up to approximately 50% of the rainfall related to cyclones.

To study these differences further, the properties of selected WCB air masses, such as potential vorticity and specific humidity, along their trajectories could be extracted from LARA. This is beyond the scope of this example, but it could provide further understanding of the factors influencing the changes in WCB occurrence during different periods.

### 4.3 Continentality of air

Continentality is generally classified by temperature: a contintental climate has at least one month averaging below 0 °C and
one above 10 °C according to the Köppen classification (Köppen, 1900). In this last science example, we show a different measure of "continentality", given by the time it takes air to reach land from the ocean. We assigned a timer to each particle that counts the time it spends over land ("travel time"). When the particle is located over land, the timer is incremented by one hour every time step; when the particle is located over the ocean, the timer is reset to zero. To make this timer representative for air-sea interactions, we reset it over the ocean only if the particle is also located in the lowest 1000 m (very roughly
corresponding to the atmospheric boundary layer height).

In this way we tracked all 6 million particles over the period 2011-2016. As shown in Figure 4, for particles residing in the lowest 1000 m above ground, the resulting travel times are high in Asia and North America, where continental climates are typically found, but also in Africa and South America over the Andes. The highest values are located north of the Tibetan plateau, in a climate that is classified as arid desert (Beck et al., 2023). Generally, continental climates are preferentially found
at high latitudes according to the Köppen classification (Beck et al., 2023), whereas our classification also results in long residence times in subtropical desert regions. This shows that the Eulerian measure of continentality (temperature variability) does not always agree with the Lagrangian measure (travel time over land).

This example could be further explored to measure differences related to seasonality or climate change. For example, the boundary layer height could be used to reset the timer, and better agreement with more classical definitions of continentality





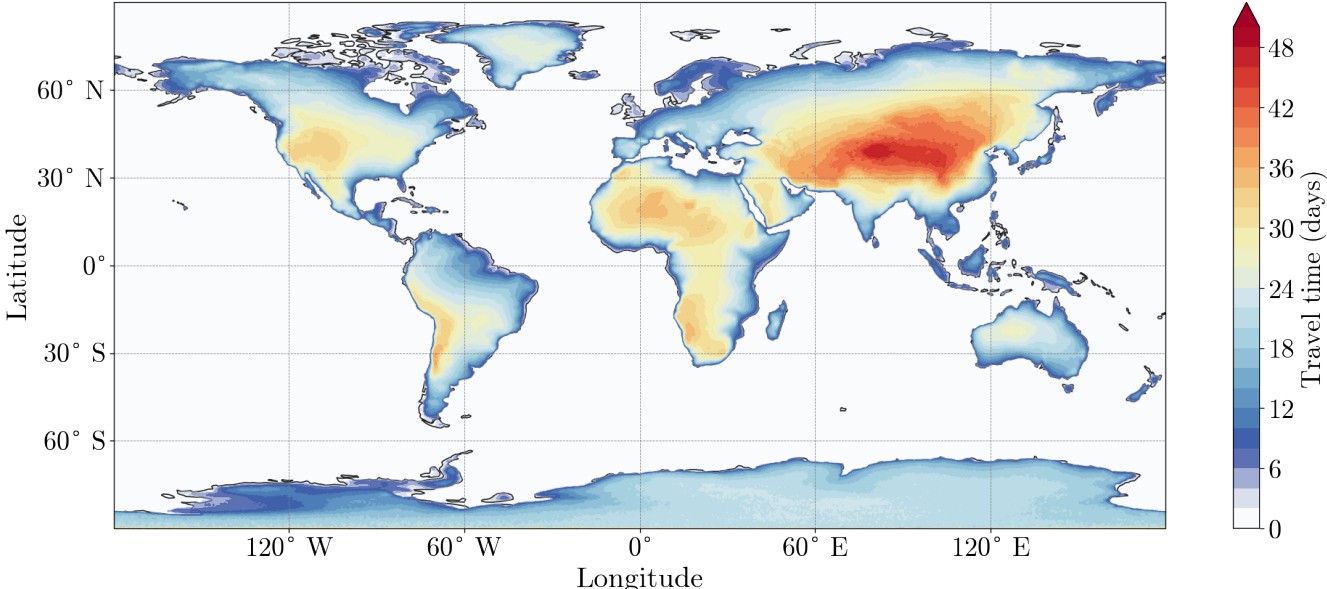

**Figure 4.** Lagrangian continentality of air. The colors show how many days particles have spent over land or above 1000 m before reaching the respective location. Only particles arriving in the lowest 1000 m are shown.

could perhaps be obtained by resetting the particle timer only over open water and not over sea ice. Furthermore, moisture tracking could be used to quantify the deposition of oceanic water over the continent by using a moisture tracking diagnostic, such as WaterSip (Sodemann et al., 2008; Fremme and Sodemann, 2019).

### 4.4 Quantifying assimilation errors

The quality of trajectories within LARA is susceptible to interpolation inaccuracies, numerical errors and uncertainties in the
turbulence and convection parameterisations, which all in turn result in inaccuracies in the vertical density distribution as shown in section 3.1. Moreover, since LARA uses wind velocities from the ERA5 reanalysis, the quality of LARA is directly linked to the quality of the ERA5 data, as shown by, e.g., Arnold et al. (2015) who investigated the impact of driving FLEXPART with various datasets. One aspect of the ERA5 data quality is the dynamical consistency of subsequent meteorological fields. Due to the assimilation of observations into ECMWF's IFS model in separate 12-hour time windows, dynamical inconsistencies
in time series of the ERA5 data occur especially when switching from one data assimilation interval to the next, resulting in discontinuities in the trajectories. Data assimilation uses available observations and is therefore dependent on both the quality and quantity of such observations (Bell et al., 2021). In addition, different meteorological fields are affected more or less strongly by assimilating different types of observations and therefore the quality of one field does not necessarily guarantee that of others. Due to changes in the observation system, there are also large changes in the quality of the ERA5 data over time.
In this final example, we present a methodology for diagnosing the impact of data assimilation on the conservation of semi-

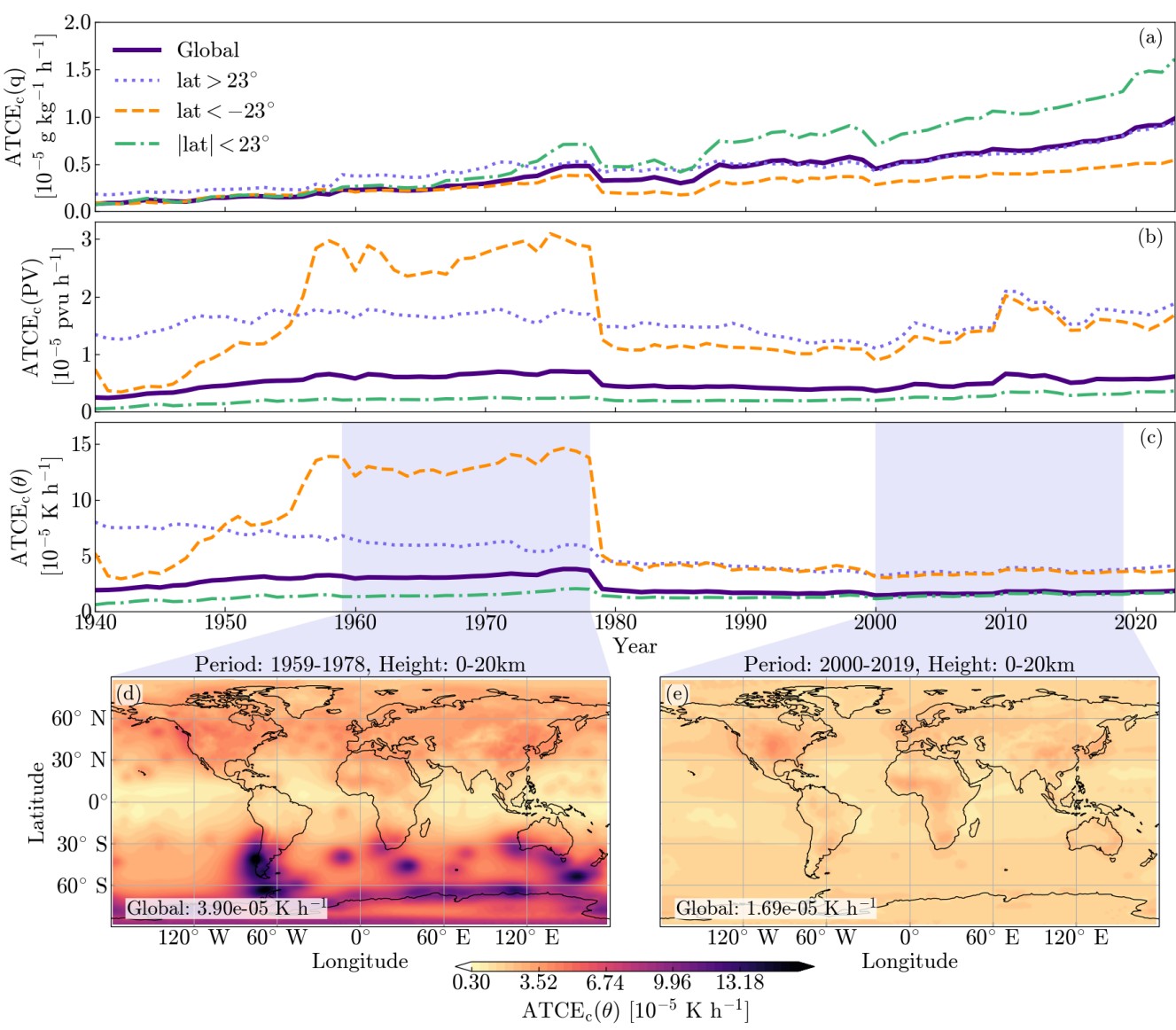

**Figure 5.** Mean annual absolute baseline-corrected tracer conservation error (ATCE$_c$) of specific humidity (**a**), potential vorticity (**b**), and potential temperature (**c**). Assimilation errors are calculated for particles at altitudes below 20 km using Eq. 3 over the hours when 12-hour assimilation windows change. Results are presented with a baseline subtracted, defined as the mean ATCE of the hours preceding and following the hour when the assimilation window change occurs. The total ATCE (thick purple line) is decomposed into latitudinal regions, namely the Northern Hemisphere north of 23° N (blue dotted line), the Southern Hemisphere south of -23° N (yellow dashed line), and the tropics between an absolute latitude of 23 ° N (green dot-dashed line). The mean ATCE$_c$ of potential temperature is mapped on a 2° by 2° grid for the period 1959-1978 (**d**) and for the period 2000-2019 (**e**). Average values are presented in the lower left of each panel.

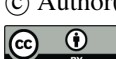


conserved tracers along the trajectories throughout the entire ERA5 period. For this, we consider specific humidity, potential vorticity, and potential temperature. These quantities are affected by different physical processes (radiative heating/cooling, evaporation, etc.) and so are not perfectly conserved in the atmosphere. However, a decent degree of conservation is expected over short time intervals in regions of the atmosphere where diabatic processes are rather slow (i.e., in clear air outside the

atmospheric boundary layer). Therefore, changes in the degree of conservation of these semi-conserved parameters over time can be used as an indication of the impact of changes in the observation system on the dynamical consistency of the ERA5 dataset.

The ERA5 data assimilation time windows are 12 hours long and steps between these windows occur every day at 09:00 and 21:00 UTC. We measure the change within specific humidity, potential vorticity, and potential temperature of all particles

between the start of each step, $t_j$ (at 09:00 and 21:00 each day) and the following hour, $t_{j+1}$ (at 10:00 and 22:00 each day) that are below 20 km. This gives us the absolute tracer conservation errors (ATCEs) (Stohl et al., 1998b) during assimilation time steps:

$$ATCE_a(\gamma) = \frac{1}{N_a N_{\text{part}}} \sum_{j=1}^{N_a} \sum_{i=1}^{N_{\text{part}}} |\gamma_i(t_{j+1}) - \gamma_i(t_j)|, \tag{1}$$

where $\gamma_i$ is the semi-conserved property of particle $i$ at time $t_j$, $N_{\text{part}}$ is the total number of particles considered, and $N_a$ are

the total number of assimilation time steps in the selected time range. As mentioned above, these properties are not perfectly conserved in the atmosphere, given that air masses are subject to a multitude of physical processes resulting in changes. Furthermore, errors in trajectory calculation other than those resulting from data assimilation as well as the interpolation of the semi-conserved quantities themselves produce conservation errors. Therefore, to measure the degree of trajectory inaccuracy resulting from data assimilation in ERA5, it is necessary to establish a baseline. We define this baseline as the average of the

hour before and after the assimilation time step:

$$ATCE_b(\gamma) = \frac{1}{2N_a N_{\text{part}}} \sum_{j=1}^{N_a} \sum_{i=1}^{N_{\text{part}}} |\gamma_i(t_{j+2}) - \gamma_i(t_{j+1})| + |\gamma_i(t_j) - \gamma_i(t_{j-1})|, \tag{2}$$

We note that the $ATCE_b$ values are always smaller than the $ATCE_a$ values and show no substantial temporal trends. We therefore measure the effect of the data assimilation on the dynamical consistency of meteorological fields by subtracting the baseline conservation error from the conservation error during the data assimilation:

$$ATCE_c(\gamma) = ATCE_a(\gamma) - ATCE_b(\gamma) \tag{3}$$

The resulting $ATCE_c$ values show substantial changes over time. Figure 5 shows both the full 84-year period trend of $ATCE_c$ for specific humidity, potential vorticity and potential temperature, as well as the spatial global distribution of $ATCE_c$ values for potential temperature. Assimilation discontinuities are generally the lowest in the early 1940s, when observations were scarce and the IFS model was only weakly constrained. They then grow until 1978, when progressively more observations

were assimilated. However, striking is the steep drop in tracer conservation discontinuities after 1978. For potential vorticity and potential temperature, this drop originates mostly from changes in the Southern Hemisphere, where the largest increase





in observation data occurred. This is in line with the introduction of assimilation of data from the TOVS (TIROS Operational Vertical Sounder) satellite series at the end of 1978 (Bell et al., 2021). An increase in the quantity of observational data, and thus a stronger observational constraint, therefore not necessarily causes a degradation in tracer conservation. We have highlighted

the period prior to this drop, when discontinuities due to data assimilation were at their highest, by mapping the potential temperature discontinuities on a 2° by 2° grid (Fig. 5d). Assimilation discontinuities in the southern hemisphere are strongly concentrated in certain regions and are, overall, much larger than in the northern hemisphere in the period before 1979 (Fig. 5d), even though before 1979 data availability in the southern hemisphere was much more sparse than in the northern hemisphere (Soci et al., 2024). It seems that assimilation of the few localized observations in the Southern Hemisphere, as visible in Fig. 1

of Soci et al. (2024), caused large changes in the model state. This is particularly evident over South America and the Antarctic Peninsula, where land-based observations were available. In the northern hemisphere, $ATCE_c$ values for potential temperature are more evenly distributed over the whole domain. After 1979, the situation becomes similar in the southern hemisphere as visible in Fig. 5e. It is also interesting to see that $ATCE_c$ values for specific humidity increase throughout the ERA5 period, while $ATCE_c$ values for potential temperature remained nearly constant after 1979.

Similar results to above could be obtained by analysing the change in the gridded ERA5 data before and after assimilation timesteps directly, but this would not include the information *along* the trajectories of air masses through space. Expanding on this example, it would be possible to trace particles selected from high-assimilation discontinuity regions and investigate which downstream regions are strongly affected by "shocks" induced by the data assimilation. Similarly, it would be possible to trace regions with known discrepancies in meteorological parameters backward in time to investigate where and when the affected

air masses underwent strong changes in meteorological properties due to the data assimilation. Such studies could help identify problems in the data assimilation system.

## 5   Conclusions

This study presents LARA, a Lagrangian reanalysis dataset, and outlines its potential and limitations in studying atmospheric transport processes. The dataset was created using the latest version of FLEXPART (Bakels et al., 2024) with meteorological

input data from the ERA5 reanalysis (Hersbach et al., 2020), and contains the hourly instantaneous and averaged values of positions, potential vorticity, specific humidity, density, temperature, and pressure for six million particles over the past 84 years. If users require other meteorological variables, they could be interpolated, e.g., from gridded ERA5 data, to the particle positions in a post-processing step. The full dataset is divided in parts of eight years with a full year overlap between the periods. Within each eight-year period, particle trajectories are continuous and particles are distributed homogeneously in the

whole atmosphere. We show that the particle distribution in the troposphere approximately fulfills the well-mixed condition, i.e. the distribution of particles in the troposphere shows relatively small deviations from the distribution of air density throughout the whole period. Small deviations of $< 6.6\%$ indicate potential for further improvements but should not limit the usability of the LARA dataset for most conceivable applications. However, above the troposphere the errors are larger, and caution should to be taken when using LARA for stratospheric studies.





We provided four different examples, with the aim of demonstrating the wide range of applications for which LARA can be used. By selecting particles that cross the equator, and following these for 20 days forward, we used their distribution after 20 days to establish a measure of the Hadley cell transport. This revealed a strong link between the latitudinal extension of the particle distribution and the PDO index. In a similar way, particles could be selected and traced within the past 84 years given any location. In our second example, the warm conveyor belt analysis, we showed the possibility of using LARA to
find coherent air streams on the basis of dynamical trajectory criteria. We found shifts in the occurrence and intensity of these events over time, with implications for understanding the impact of climate variability on extreme weather events. Similarly, properties of the particles within WCBs could be analysed to better understand, for example, the relation between WCBs and potential vorticity. However, we caution users that the quality of LARA is determined by the underlying ERA5 data, and that care must to be taken when studying trends over periods that use different assimilation data. In our third example, we used
LARA to obtain a Lagrangian definition of continentality. By tracking the time particles spend over land, we quantified how long it takes air to reach land from the ocean. This analysis could be expanded by additionally tracking moisture, e.g., using a moisture source diagnostic. In our last example, we showed that the LARA dataset can be used to diagnose properties in the ERA5 data. We highlighted how changes in the observation system and data assimilation procedures introduce dynamical inconsistencies that can be quantified using the LARA dataset. By examining the conservation of semi-conserved tracers,
such as specific humidity, potential vorticity, and potential temperature, this study provides a means to assess the reliability of LARA data across different periods. It should be noted that the analytical purposes presented here are not intended to be all-encompassing.

Perhaps the largest limitation of the LARA data set is the modest number of particles tracked. This limits the effective time-space resolution of analyses, especially when it comes to case studies of individual events (e.g., extreme precipitation events).
Perhaps LARA can trigger enough interest in the scientific community that future Lagrangian reanalyses will be carried out by meteorological centers with larger computational resources than those available at a university. This will facilitate the tracking and archiving of a much larger number of particles, and could include a more direct connection to the Eulerian reanalysis, facilitating the efficient provision of additional meteorological variables along the particle trajectories from within the same data archive.

*Code and data availability.* The dataset is hosted by the Earth Observation Data Center (EODC) and available to the public via: https://data.eodc.eu/collections/LARA/ as a zarr dataset. LARA is produced using FLEXPART, available from https://doi.org/10.5281/zenodo.12706632 (Bakels et al., 2024). All scripts necessary to create the dataset, as well as the scripts to reproduce the analyses and plots in this work are provided in the Supplement. In addition, processed LARA data that were used to create the plots can be found here: https://doi.org/10.5281/zenodo.14639472.



*Author contributions.*   LB created the dataset; validated the dataset; created most figures; and wrote the majority of the paper. MB managed the conversion to zarr and the storage of the dataset; and provided text. MD, in collaboration with GB, created the section on the continentality of air; provided feedback and contributed to the text. AP contributed ideas, tested the dataset, and helped with the design of plots and text. VL contributed to the analysis of the validation and limitation of the dataset. GB, in collaboration with MD, analysed the continentality of air using the LARA dataset. LH made hosting arrangement of the data and contributed textual feedback. AS provided ideas, feedback on the

analysis, and contributed to the text.

*Competing interests.*   No competing interests

*Acknowledgements.*   We acknowledge valuable input from Michael Meyer, Sabine Eckhardt, and Katharina Baier. We acknowledge the Earth Observation Data Center (EODC) for storing and distributing the data using their facilities. This study has been supported by the Dr. Gottfried and Dr. Vera Weiss Science Foundation and the Austrian Science Fund in the framework of the project no. P 34170-N, "A demonstration of

a Lagrangian re-analysis (LARA)". The computational results presented have been achieved using the Vienna Scientific Cluster (VSC). In addition, we acknowledge various public Python packages that have benefited our study: NumPy (Harris et al., 2020), Matplotlib (Hunter, 2007), Xarray (Hoyer and Hamman, 2017), SciPy (Virtanen et al., 2020), and cartopy (Met Office, 2010 - 2015).



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
