# Peer review of "LARA: a Lagrangian Reanalysis based on ERA5 spanning from 1940 to 2023"

_Earth System Science Data, 2025_

## Referee Comment (RC2)

Review of paper

**LARA: a Lagrangian Reanalysis based on ERA5 spanning from 1940 to 2023**

by L. Bakels et al.

submitted to ESSD

This is an excellent paper about a very interesting open-access dataset (LARA). The authors offer a state-of-the-art global Lagrangian climatology to the research community, which they calculated using FLEXPART and more than 80 years of ERA5 reanalyses. For the first time, such a comprehensive and valuable dataset can be used by researchers worldwide – and potential applications are numerous. The authors provide a selection of interesting example applications that nicely illustrate the potential of LARA. The paper is generally well written, but a few clarifications can further improve the paper. My main concern is that the first application is not at all easy to understand (the way it is presented at the moment) and this might give a wrong impression about the usability of Lagrangian diagnostics. Overall, I recommend accepting this excellent paper with minor revisions.

**Comments (most of them are minor)**

- 1) L17: "by time" sounds strange, maybe "as the time" works better.
- 2) L54: we typically write "airstreams" as one word
- 3) L69: should read "ERA-Interim"
- 4) Table 1: units should be "m s-1" instead of "m/s"; symbols should be in math mode, e.g., *u*, *T*, *q*; not sure why you use rather unusual variable names like "sp", "2t" etc.; *ps*, *T2m* would be more common → consider using more standard variable names
- 5) L123: it would be interesting to have a bit more information about the 6 million particles: how did you decide for this number? What is the resulting mass represented by each particle? How many particles are on average in a 1x1 deg column?
- 6) L126: a bit strange "the full period ... would take ... to complete", maybe better "the calculations for the full period ..."
- 7) L128/129: confusing, first the overlap period is one year, then 3 months only?
- L132: FYI: the 300 s (= 5 min) time step correspond to 1/12 of the data input interval (1 h), which is the default approach used in LAGRANTO
- 9) L141: the periods are 8-y long

- 10) L148: I assume that BL height is directly from ERA5; what about the tropopause height? Which definition is used to calculate the tropopause height?
- 11)L149-151: I don't understand these sentences, why are BL and TP height not simply interpolated to the horizontal position of the air parcels?
- 12) L160: I don't understand "a month's worth of files normalised per variable"
- 13) L172: difficult to understand what these Spearman coefficients refer to (particle positions or tracer concentrations?)
- 14) L210: in line with my comment 5: please provide more information about the density of parcels typically available in a box of, e.g., 100 km x 100 km x 100 hPa
- 15) L220: "or are fully explored on themselves" sounds strange, I suggest something like ", nor are they investigated in full detail here"
- 16) L227: "periods that use different assimilation data" maybe this aspect could be discussed briefly in Sect. 2.1. The ERA5 dataset is as consistent as it can be, but there are still issues with changes in the datasets available for assimulation.
- 17) L238: "The left panel of figure 2"  $\rightarrow$  please use panel lables and write "Figure 2a ..."
- 18) L241: "within the lower parts of the lower atmosphere" is not very clear, can you give a pressure or height range?
- 19) L243: "in panel (a2)" should maybe read "in Fig. 2a2" (however quite unusual), or you better change panel labels to (a, b, ... j)
- 20) L243-266: honestly, I am a bit lost with the text and the figure ... all rather complicated. Do you need so many panels to make the main point? Do you need the correlations with AMO, PDO, ...? If yes, then the reader requires a more careful explanation of what is shown and why and how to interpret the results. It would be a pity if the first application of LARA was so complicated that readers get the impression that Lagrangian investigations are hard to understand.
- 21) L285: 500 hPa should read 600 hPa
- 22) L290: I don't understand the criterion "air mass within a WCB per square metre to be larger than 1 kg", is this a criterion to guarantee a certain density of particles that fulfill the WCB criterion of ascent?
- 23) L293: maybe worth noting that your WCB climatology uses different units than, e.g., Madonna et al. if I interpret your unit correctly, then it refers to a vertical mass flux in WCBs (kg /  $m^2$

/ s). The comparison with WCB frequencies (as in Madonna et al.) is therefore qualitative (which is fine, just maybe helping the readers to get along with the different units and values)

- 24) L321: I very much liked this application; it is easy to understand and clearly demonstrates something that could not been obtained by Eulerian analyses
- 25) L344: this could be better mentioned already in Sect. 3.1
- 26) L388: it is a very interesting results that "... ATCEc values for specific humidity increase throughout the ERA5 period" do you have a hypothesis why this is the case?
- 27) L390: "Similar results to above" sounds strange, maybe "to the ones discussed above"

---

## Author Comment (AC1)

**Reviewer 1:**

We sincerely thank the reviewer for taking the time to thoroughly review our manuscript. The detailed feedback and constructive suggestions are very appreciated. We believe to have made the necessary revisions to address all comments.

In the following, responses are in blue, and quoted text is shown in green. Text after the little arrow ' $\rightarrow$ ' is newly introduced or modified manuscript text in the reaction to the reviewer's comments.

**General comments**

This paper introduces LARA (Lagrangian Reanalysis), a novel global dataset created by converting ERA5 reanalysis data into a Lagrangian format using the FLEXPART model. Unlike traditional Eulerian reanalyses, LARA tracks six million air particles from 1940 to 2023, providing high-resolution, hourly data on their positions and atmospheric properties. This dataset supports studies on atmospheric transport, energy and moisture fluxes, and extreme events. It is validated for internal consistency. LARA is available in Zarr data format, accompanied by analysis scripts and four illustrative applications, such as Hadley cell evolution and diagnostics of data assimilation discontinuities in ERA5. Overall, LARA appears to be a powerful tool for Lagrangian-based atmospheric research.

The methodology of the study is sound. FLEXPART is a reliable model, and the transition to a Lagrangian perspective is well-justified. The validation is appropriate, and the analysis is carefully executed, offering a valuable dataset with wide applications.

One general comment that remains open is how the number of six million air parcels used in LARA was determined. This number is briefly discussed in the conclusion but seems based on computational and storage constraints rather than a sensitivity analysis? Clarifying whether tests were conducted to assess the adequacy of this particle count would strengthen the manuscript. It would also be useful to discuss whether scientific results in Section 4 would remain robust with different particle counts.

It is indeed constrained by mainly the storage. The number of particles sets a lower limit on the volume in time and space needed to sample enough particles to obtain statistically significant results. This is described in Section 3.2. Since all our examples use large volumes, these results will not be affected by increasing the particle number, although it would make some of the maps shown smoother and possibly reduce some noise. We added the following statement to Section 2.3:

The dataset is large, with a total size of approximately 320 TB, and making it openly available constrained our choice of total particle number and temporal output resolution. Section 3.2 describes possible limitations on the use of the dataset due to its resolution.

Overall, the manuscript is well written, scientifically rigorous, and uses proper terminology and referencing. It follows a clear structure and presents concepts and results in an accessible and precise manner. While some sections are detailed, this aligns with the paper's dual role as a dataset description and usage guide. Overall, the manuscript meets high standards of clarity, conciseness, and scientific communication. I recommend publication subject to addressing the specific comments and technical corrections listed below.

**Specific comments**

Lines 34-39: Several major reanalyses, including MERRA and JRA-55, are mentioned, but it would be useful to note that newer versions are available, MERRA-2 and JRA-3Q. Citing a recent reanalysis intercomparison, such as from the S-RIP/A-RIP initiatives, would provide broader context on how these datasets compare in performance and scope.

**The citations and mention of MERRA-2, JRA-3Q, and the intercomparison project are added.**

Lines 70-73: The Lagrangian datasets by Sodemann et al. and Vázquez et al. are described as 'available on request', but it's unclear how they can be accessed. If they are not openly available in a FAIR-compliant manner, it may be better to omit or clarify these statements to avoid misleading readers, especially compared to the open-access nature of LARA.

**We omitted the statements in the revised version of the paper.**

Line 74: The phrase 'showing large decrease in particle distribution degradation in the troposphere over time' is somewhat unclear. It would help to briefly explain what is meant by 'particle distribution degradation', such as whether it refers to deviations in air density, loss of particle ensemble representativeness, or numerical diffusion effects.

We added the following statement: Since then, a new version of FLEXPART (version 11) has been released, showing a large decrease in particle distribution degradation in the troposphere over time, where in version 10.4, the particle number density does not well represent air density after some time (Bakels et al. 2024).

Line 78: The manuscript refers to the 'full period of 1940–2024 ERA5 reanalysis', but the title states 1940–2023. This discrepancy should be addressed for consistency, clarifying if the dataset includes data through March 2024, as mentioned later in the paper.

**Thank you for pointing this out. We corrected the time period running until March 2024. Since we only cover 3 months of 2024, we decided to leave the title intact.**

Line 82: Instead of providing just a web link, include a proper citation with a DOI for the LARA dataset to ensure long-term accessibility and facilitate referencing, in line with best practices for data publication. At the time of this review, the web link was not accessible, unfortunately. The web site showed 'Internal Server Error'.

**Our apologies, we replaced it with the proper persistent identifier, which also includes a description of the dataset (https://phaidra.univie.ac.at/o:2121554).**

Table 1: The additional notes in the 'unit' column are unclear. For example, it's not obvious what is meant by etadot having units of  $s^{-1}$  but 'internally: Pa  $s^{-1}$ '. If these notes are crucial, move them to footnotes beneath the table for better clarity without cluttering the unit column.

**Thank you for the suggestion. The notes have been moved to footnotes.**

Line 93: The manuscript mentions using ERA5 data at  $0.5^{\circ} \times 0.5^{\circ}$  resolution, although ERA5 is (natively) available at  $0.25^{\circ} \times 0.25^{\circ}$ . Clarifying why the reduced resolution was chosen (e.g., due to computational or storage constraints) would help improve transparency regarding the dataset's accuracy.

"For the creation of the LARA dataset, we used the most recent reanalysis dataset of ECMWF, ERA5, with hourly 0.5x0.5 due to storage constraints data as input on 137 vertical model levels, of which 88 are located below 20 km."  $\rightarrow$  "..., with its native hourly temporal and 137-level vertical resolution and a reduced horizontal resolution of 0.5x0.5 due to storage constraints."

Line 121: A brief explanation of the 'domain-filling option' would help readers unfamiliar with the concept. Clarifying that it initializes particles to uniformly represent the entire atmospheric mass and discussing its relevance to mixing and atmospheric transport would enhance clarity.

The following clarification has been added: "..., which discretises the full atmosphere into particles of equal mass. The particles are initialized such that their number density is proportional to the density of air, and each of them represents an equal fraction of the total atmospheric mass. "

Lines 126–128: Since the data processing was split into overlapping streams, clarify if consistency checks were done across stream boundaries. It would be useful to mention whether users need to account for potential artifacts or discontinuities when working across different stream periods.

Our apologies, there were some mistakes in the text that are now clarified. There is a full year overlap, since particles cannot be traced between the different streams. The full year of overlap guarantees that particles can always be traced - also across the boundaries - below the span of one year (corrected in the text), which is more than enough for most applications. The difference in 'quality' of the overlapping years at the start and the end of the stream is shown in section 3.1.

Lines 126–130: The explanation of overlapping periods between computational streams is unclear. The text first states 'one full year for each period', then mentions a 'three-month overlap', which is confusing. Please clarify whether the overlap is one year or three months, and explain how it ensures continuity in trajectory tracking.

Apologies, this is corrected and clarified now, see above.

Lines 132–136: While the impact of time step length on trajectory accuracy is discussed, the numerical integration scheme used in FLEXPART is not mentioned. Specifying the integration method (e.g., Runge-Kutta, Euler) would provide a more complete understanding of the model setup and accuracy.

The following statement is added to the FLEXPART section 2.2: "FLEXPART uses a first order Euler method in addition to a Petterssen correction (Petterssen, 1940) for advancing particles."

Lines 139–141: The inclusion of compiler flags and OpenMP settings may be overly detailed for an ESSD paper. Unless these settings impact the results or reproducibility, consider omitting it for clarity and focus.

**The flags have been omitted.**

Lines 144–145: The dataset provides hourly means for several variables, but adding additional statistics like minimum, maximum, or standard deviation could be valuable, especially for applications involving extremes or uncertainty. If not included, note whether these could be added in future versions.

We considered adding such values, but due to storage constraints decided against it. However, some variability could be derived from comparing the instantaneous values to the averaged values. Uncertainties could be derived from comparing groups of particles to each other given the same volume, and maxima and minima should emerge when using large enough numbers of instantaneous particle values.

Lines 157–161: Include an estimate of the total size (e.g., in terabytes) of the full LARA dataset, especially since storage format and compression are discussed. This would help users understand data management requirements for downloading or processing large subsets.

**A statement has been added (see first comment).**

Line 170: Perhaps briefly elaborate on improvements to the FLEXPART interpolation scheme in the latest version. Summarize enhancements to help readers understand their role in reducing potential errors.

We added the following clarification: "Using the improved interpolation scheme of FLEXPART 11, a reduction of ~15% and ~8% in PV errors is observed above 10 km and between 5-10 km, respectively."  $\rightarrow$  "Using the improved interpolation scheme of FLEXPART 11, which removed an interpolation step by advancing particles using the native vertical coordinate grid of the ERA5 data, a reduction of ~15% and ~8% in PV errors is observed above 10 km and between 5-10 km, respectively." We also added: "More details on FLEXPART 11 improvements can be found in Bakels et al. (2024)."

Figure 1: The x-axis label refers to absolute differences,  $|\rho_{part} - \rho_{ERA5}|$ , but negative values suggest signed differences ( $\rho_{part} - \rho_{ERA5}$ ) are shown. Update the label to reflect this. Also, suggest to change the y-axis label to 'Geopotential height (m)' for clarity.

**This has been corrected, thank you.**

Lines 193-194: The sentence about maximum deviation of 0.6% from latitudinal-averaged volume is unclear. Clarify what is meant by 'latitudinal-averaged volume' and how the 0.6% deviation is calculated.

This is a mistake, it is the deviation over the whole latitudinal volume, with longitudes spaced by 0.5 degrees. We decided to remove the paragraph: "The maximum deviation ... can be larger." since it does not provide any new information and potentially causes confusion instead.

Line 199: Suggest to rephrase 'it is always better to take the air density from the \_ERA5\_ values interpolated to the particle positions' for clarity.

**Rephrased as suggested.**

Lines 201-202: The sentence about calculating mass using average air density within the source or receptor volume is unclear. Does this affect mass conservation? Should users always rely on air density or mass derived from ERA5 data at the particle locations, rather than using the mass of Lagrangian particles?

We removed this sentence and the next ("Similarly, when the total transported mass ... source or receptor volume."), since they cause more confusion than they are helpful. Figure 1 should be sufficient for explaining the particle density discrepancy.

Figure 2: The figure seems somewhat overloaded with information. Consider splitting it into two figures for panels a and b or removing unnecessary curves/data to improve clarity.

The figure has been simplified, removing unnecessary curves, while also putting more emphasis on explaining the methods. We separated the figure into three steps: 1) maps of the selected particles; 2) Particle distribution and PDFs; and 3) an analysis that could be conducted using these distributions. We hope this makes the figure more valuable in a pedagogical way.

Lines 263-266 and 273-276: The phrasing could be adjusted to avoid casting doubt on previous results. If only local/regional analysis is meaningful, why focus on global results in Fig. 2? A more confident presentation of the findings could strengthen this section.

Thank you for pointing this out. We have rephrased the paragraph in a more positive light, explaining why we do not go in much detail for the examples: "To demonstrate the potential of LARA, we only present global results, but it would be interesting to extend the analysis to particular regions (i.e. Pacific, Atlantic, Indian Ocean). This would then allow for the investigation of other processes, e.g. global warming, ...".

Lines 299-300: Could convection parametrizations influence the results?

This is unlikely. Eckhardt et al. (2004) did not use a convection scheme for their trajectory calculations, while here we did, but the results are very similar. The text read as if convection parameterisation was applied to Eckhardt et al. (2004), which is incorrect, and we therefore added the following clarification to the text: **"Eckhardt et al. (2004) did not use a model that parameterises convection, either. However, when combined with** the requirement that the air mass within a WCB be greater than 1 kg per square metre - which implies a certain degree of coherence in airstream motion - **it reasonably corrects for the erroneous classification of deep convection.**"

Lines 318-320: Why use a 1000 m proxy for boundary layer height, given that the actual boundary layer height is available in the LARA dataset?

**We agree it might have been better to use the actual boundary layer height. A more in-depth follow-on analysis (Brack et al., in preparation) actually used boundary layer heights. For the purpose of demonstration, however, we consider the use of a constant height sufficient.**

Figure 5: The tracer conservation errors are averaged up to 20 km altitude, but tracers considered here vary exponentially with height, whereas particle numbers decrease exponentially with height. Is there any significant altitude dependence in tracer conservation errors? It could be insightful to compare conservation errors across different layers (e.g., lower, middle, upper troposphere, and lower stratosphere).

This would indeed be very interesting. As earlier shown in the FLEXPART version 11 paper (Bakels et al. 2024), the altitude makes a large difference in conservation error. However, our aim for this paper is to show possible avenues of what could be done with the dataset, without going into much detail. Of course, we hope these questions will be picked up by others and be further explored upon.

Line 367: Add '(not shown)' since this is not demonstrated in Fig. 5.

**Added "(not shown)".**

Lines 377-378: The improvement in consistency may be due to the broader onset of the modern satellite era, not just the TOVS instrument.

We improved the statement including the broader onset of the modern satellite era by replacing the following: "This is in line with the introduction of assimilation of data from the TOVS (TIROS Operational Vertical Sounder) satellite series at the end of 1978."  $\rightarrow$  "This is in line with the introduction of assimilation **of satellite data, such as from the** TOVS (TIROS Operational Vertical Sounder) satellite series at the end of 1978."

**Technical corrections**

Please ensure consistent capitalization and abbreviations for terms like 'Section' and 'Figure' throughout the text, following Copernicus manuscript guidelines.

Thank you, this has now been corrected.

Line 69: rephrase 'ERAInterim' as 'ERA-Interim'

**Corrected.**

Line 83: please check and standardize the spelling of 'Zarr' data format throughout the manuscript

**It has been standardised now.**

Line 133: 'deviate by approximately 20% less from ERA5 air densities' – Do you mean '20% or less'?

We clarified it by replacing the sentence as follows: "We found that particle densities deviate approximately 20% less from ERA5 air densities when using 300 seconds time steps compared to 600 second time steps."  $\rightarrow$  "We found that particle densities show smaller deviations, with a reduction of the difference of particle mass densities to ERA5 air densities of approximately 20% when using 300 seconds time steps compared to 600 second time steps."  $\rightarrow$  "We found that particle densities show smaller deviations, with a reduction of the difference of particle mass densities to ERA5 air densities of approximately 20% when using 300 seconds time steps compared to 600 second time steps."

Line 141: Do you mean '12 eight-year periods' (considering the overlap)?

Thank you, we corrected the mistake.

Table 2: Column header 'unit' -> 'Unit'. Fix unit 'K' for tropopause height.

Thank you, it is corrected.

Line 254: The acronym 'ERSSTV5' is not introduced.

We added the full name (Extended Reconstructed Sea Surface Temperature version 5).

Line 285: model used in \_that\_ study

**Corrected**

Figure 5: suggest to replace '-23°N' by '23°S' in the caption

**Corrected**

Lines 382-383: Please check the capitalization of 'Northern Hemisphere' and 'Southern Hemisphere' throughout the manuscript for consistency.

We have now consistently capitalised it.

---

## Author Comment (AC2)

**Reviewer 2:**

This is an excellent paper about a very interesting open-access dataset (LARA). The authors offer a state-of-the-art global Lagrangian climatology to the research community, which they calculated using FLEXPART and more than 80 years of ERA5 reanalyses. For the first time, such a comprehensive and valuable dataset can be used by researchers worldwide – and potential applications are numerous. The authors provide a selection of interesting example applications that nicely illustrate the potential of LARA. The paper is generally well written, but a few clarifications can further improve the paper. My main concern is that the first application is not at all easy to understand (the way it is presented at the moment) and this might give a wrong impression about the usability of Lagrangian diagnostics. Overall, I recommend accepting this excellent paper with minor revisions.

We thank the reviewer for the very positive evaluation of our manuscript, for the detailed feedback and constructive suggestions. We have carefully considered all comments and have made the necessary revisions to address the points that were raised. In the following, responses are in blue, and quoted text is shown in green. Text after the little arrow ' $\rightarrow$ ' is newly introduced or modified manuscript text in the reaction to the reviewer's comments.

**Comments (most of them are minor)**

1) L17: "by time" sounds strange, maybe "as the time" works better.

We modified the sentence as follows: "a measure of continentality by time it takes air to reach land from the ocean"  $\rightarrow$  "a measure of continentality **based on the** time it takes for air to reach land from the ocean"

2) L54: we typically write "airstreams" as one word

Thank you, this has been corrected.

3) L69: should read "ERA-Interim"

Thank you, we corrected this

4) Table 1: units should be "m s-1" instead of "m/s"; symbols should be in math mode, e.g., u, T, q; not sure why you use rather unusual variable names like "sp", "2t" etc.; ps, T2m would be more common à consider using more standard variable names

Thank you, we corrected the unit. The unusual variable names are the short names given of the variables of the ERA5 dataset. We clarified this by replacing the header "Short name" with "ERA5 short name".

5) L123: it would be interesting to have a bit more information about the 6 million particles: how did you decide for this number? What is the resulting mass represented by each particle? How many particles are on average in a 1x1 deg column?

A similar concern was voiced by Reviewer 1. We added a statement on why we chose 6 million particles (storage constraints, 320TB). We also added a short explanation about the domain-filling option which answers the question about the mass of each particle (= total atmospheric mass divided by 6 million). The number of particles per grid cell volume varies

greatly across the globe, due to the smaller grid cell volumes closer to the poles compared to the equator. We added the following to the text: "... fixed air mass of approximately 0.86x10^12 kg of air, ..." and "A 1°x1° column at the equator contains approximately 300 particles at any point in time."

6) L126: a bit strange "the full period ... would take ... to complete", maybe better "the calculations for the full period ..."

We agree and took the suggestion

7) L128/129: confusing, first the overlap period is one year, then 3 months only?

This is a mistake and has been corrected (it should be 1 full year).

8) L132: FYI: the 300 s (= 5 min) time step correspond to 1/12 of the data input interval (1 h), which is the default approach used in LAGRANTO

LAGRANTO also has an iterative Petterssen correction, making the integration more correct. On the other hand, as far as we are aware, it also does not parameterise convection and turbulence, which should result in much less stochastic noise as compared to FLEXPART. For these reasons, convergence could be reached for different time step intervals. In addition, the input resolution could also be affecting the convergence point.

9) L141: the periods are 8-y long

Thank you, this has been corrected.

10) L148: I assume that BL height is directly from ERA5; what about the tropopause height? Which definition is used to calculate the tropopause height?

For clarification, the following has been added to the paragraph: "The topography is taken from ERA5, the atmospheric boundary layer height is computed with the method of Vogelezang and Holtslag (1996) based on the critical Richardson number, and the tropopause height is defined as the first stable layer to fulfill the thermal tropopause criterion (i.e. the vertical temperature gradient is smaller than 0.002 K km-1)."

Vogelezang, D. H. P. and Holtslag, A. A. M.: Evaluation and model impacts of alternative boundary-layer height formulations, Bound.-Layer Met., 81, 245–269, 1996.

11) L149-151: I don't understand these sentences, why are BL and TP height not simply interpolated to the horizontal position of the air parcels?

Since these values have only two spatial dimensions, to be space efficient, we decided to keep those in gridded form. We added our reasoning for doing so: "Therefore, to conserve storage space, these are saved as two-dimensional spatial gridded fields..."

12) L160: I don't understand "a month's worth of files normalised per variable"

The NetCDF files are organised as daily files containing all variables, while the Zarr files are organised per month and per variable, so we divided the time it took to open the NetCDF files by the number of variables to make it comparable to opening the Zarr file. This is not very relevant and we see it is confusing, so we rephrased it as follows: "...for example, opening files is ~60% faster."

13) L172: difficult to understand what these Spearman coefficients refer to (particle positions or tracer concentrations?)

The coefficient refers to tracer concentrations, which is now added to the sentence: "...with Spearman coefficients of tracer concentrations between 0.56-0.68"

14) L210: in line with my comment 5: please provide more information about the density of parcels typically available in a box of, e.g., 100 km x 100 km x 100 hPa

We added the following statement: "For example, at a single point in time, a  $1^{\circ}x1^{\circ}$  column at the equator contains approximately 300 particles, of which ~150 are below 5 km in altitude and only 10 are above 20 km."

15) L220: "or are fully explored on themselves" sounds strange, I suggest something like ", nor are they investigated in full detail here"

Thank you, we have replaced it following your suggestion.

16) L227: "periods that use different assimilation data" – maybe this aspect could be discussed briefly in Sect. 2.1. The ERA5 dataset is as consistent as it can be, but there are still issues with changes in the datasets available for assimulation.

Thank you for the suggestion. We added the following in section 2.1: "ERA5 uses a consistent model framework over the whole reanalysis period. However, the observation system has changed tremendously over this period, and thus the assimilation of observation may lead to spurious variability and trends in certain variables. Such artifacts may extend to the LARA data set, such that care should be taken when applying trend analyses."

17) L238: "The left panel of figure 2" à please use panel lables and write "Figure 2a ..."

Thank you, this has now been corrected.

18) L241: "within the lower parts of the lower atmosphere" is not very clear, can you give a pressure or height range?

By simplifying the figure and text, we removed the paragraph containing this sentence.

19) L243: "in panel (a2)" should maybe read "in Fig. 2a2" (however quite unusual), or you better change panel labels to (a, b, ... j)

The panel labels have been changed to a-h.

20) L243-266: honestly, I am a bit lost with the text and the figure ... all rather complicated. Do you need so many panels to make the main point? Do you need the correlations with AMO, PDO, ...? If yes, then the reader requires a more careful explanation of what is shown and why and how to interpret the results. It would be a pity if the first application of LARA was so complicated that readers get the impression that Lagrangian investigations are hard to understand.

We agree that Figure 2 was too complicated. The purpose of the figure is to highlight the methods of tracing particles and diagnosing trends in circulation patterns. We feel this example shows one of LARA's core strengths. To explain this better, we decided to remove panels, lines, text and added two maps of the particle selection at t=0 and t=20 days for a step-by-step explanation of the methods.

**21) L285: 500 hPa should read 600 hPa**

**It should be 500 hPa.**

22) L290: I don't understand the criterion "air mass within a WCB per square metre to be larger than 1 kg", is this a criterion to guarantee a certain density of particles that fulfill the WCB criterion of ascent?

**Indeed, it reduces noise from the convection parameterisation.**

23) L293: maybe worth noting that your WCB climatology uses different units than, e.g., Madonna et al. – if I interpret your unit correctly, then it refers to a vertical mass flux in WCBs (kg / m2/ s). The comparison with WCB frequencies (as in Madonna et al.) is therefore qualitative (which is fine, just maybe helping the readers to get along with the different units and values)

We added the following sentence: "Different units were used in earlier studies and we therefore only do a qualitative comparison."

24) L321: I very much liked this application; it is easy to understand and clearly demonstrates something that could not been obtained by Eulerian analyses

**Thank you!**

25) L344: this could be better mentioned already in Sect. 3.1

We agree and have added text in Section 2.1 (see also response to comment 16).

26) L388: it is a very interesting results that "... ATCEc values for specific humidity increase throughout the ERA5 period" – do you have a hypothesis why this is the case?

This is indeed puzzling. We do not have a certain answer for this, but hypothesise that specific humidity is a value that is more 'untouched' by assimilation in the past than in the present.

27) L390: "Similar results to above" sounds strange, maybe "to the ones discussed above".

Thank you, we adjusted the text as suggested.